# An Improved Vision Transformer Network with a Residual Convolution Block for Bamboo Resource Image Identification

**Qing Zou** [1,2,†], **Xiu Jin** [1,2,*,†] ⓘD, **Yi Song** [1,2], **Lianglong Wang** [1,2], **Shaowen Li** [1,2], **Yuan Rao** [1,2], **Xiaodan Zhang** [1,2] and **Qijuan Gao** [1,2]

1   Anhui Province Key Laboratory of Smart Agricultural Technology and Equipment, Anhui Agricultural University, Hefei 230001, China
2   College of Information and Computer Science, Anhui Agricultural University, Hefei 230001, China
*   Correspondence: jinxiu123@ahau.edu.cn
†   These authors contributed equally to this work.

**Abstract:** Bamboo is an important economic crop with up to a large number of species. The distribution of bamboo species is wide; therefore, it is difficult to collect images and make the recognition model of a bamboo species with few amount of images. In this paper, nineteen species of bamboo with a total of 3220 images are collected and divided into a training dataset, a validation dataset and a test dataset. The main structure of a residual vision transformer algorithm named ReVI is improved by combining the convolution and residual mechanisms with a vision transformer network (ViT). This experiment explores the effect of reducing the amount of bamboo training data on the performance of ReVI and ViT on the bamboo dataset. The ReVI has a better generalization of a deep model with small-scale bamboo training data than ViT. The performances of each bamboo species under the ReVI, ViT, ResNet18, VGG16, Densenet121, Xception were then compared, which showed that ReVI performed the best, with an average accuracy of 90.21%, and the reasons for the poor performance of some species are discussed. It was found that ReVI offered the efficient identification of bamboo species with few images. Therefore, the ReVI algorithm proposed in this manuscript offers the possibility of accurate and intelligent classification and recognition of bamboo resource images.

**Keywords:** bamboo resources; image classification; vision transformer; residual convolution

## 1. Introduction

As an important renewable resource, bamboo has the advantages of fast growth, short cycle and high yield and is widely used in architecture, food, furniture and landscaping [1–4]. In recent years, with the excessive destruction of the environment, the deterioration of the climate and the constant depletion of wood resources, bamboo has received much attention from society and international organizations, especially in terms of its economic and ecological value [5]. Due to the complexity of the environment in which bamboo is found and the similarity in appearance of the more than 1200 species [6], it is difficult to classify images of bamboo, and there are few bamboo experts, so it is important to assist experts in the classification and identification of bamboo. With the rapid rise and development of deep learning, image classification research based on deep learning has become increasingly extensive, and the achievements of deep learning in image classification are becoming valued and utilized by people. Bamboo classification based on a deep learning model can help experts identify different species of bamboo, which is of great significance to the protection of bamboo species diversity.

In computer vision, LeNet [7], proposed by Lecun in 1989, marked the birth of convolutional neural networks (CNNs). Alexnet [8] aroused widespread attention in the industry. VGGNET [9] focused on the effect of depth in CNNs. GooLeNet [10] began to focus on memory and computation usage and won the 2014 ILSVRC Championship with a 6.7% error rate. ResNet [11] proposed a deep residual learning architecture to

address the degradation problem, and the latest DenseNet [12] uses dense connectivity to mitigate gradient degradation, which further improves network performance. CNNs have gradually occupied the dominant position in computer vision and have been applied to an increasing number of aspects of our lives. CNNs are also more widely applied to agriculture. For example, to proactively monitor the phenological response of olive trees to changing environmental conditions, Milicevic used CNN algorithms such as VGG19 and ResNet50 to track the timing of specific surface phases in 2020, and a fivefold cross-validation classification accuracy of $0.9720 \pm 0.0057$ was produced [13]. In 2020, Quiroz proposed an image recognition method based on CNNs to detect the existence of trays with live blueberry plants [14].

Self-attention-based architectures, in particular transformers [15], have become the preferred model of choice in natural language processing (NLP). A transformer model is typically pretrained on a large corpus and then fine-tuned for specific tasks [16–18]. BERT [19] uses noise-reduction self-supervised pretraining tasks, and GPT [20,21] uses language modeling as its pretraining task. The remarkable achievements of transformers in NLP have led some researchers to propose the application of transformers to computer vision, proposing the vision transformer (ViT) model [22]. ViT applies a pure transformer network directly to an image with as few modifications as possible. When trained on medium-sized datasets (such as ImageNet), if there is no strong regularization, the accuracy of ViT will be several percentage points lower than that of ResNet of the same size. Some studies [22] show that ViT lacks some inductive biases inherent to CNNs, such as translation equivariance and locality. Therefore, ViT cannot be well summarized when the amount of data is insufficient. However, when trained on a larger dataset (14–300 M images), large-scale data training is better than inductive bias. When pretrained on the ImageNet-21k dataset or internal JFT-300M dataset, ViT approaches or exceeds the latest level on many image recognition benchmarks. The accuracy of the best model reaches 88.55% on ImageNet, 90.72% on ImageNet ReaL and 94.55% on CIFAR-100. From the performance of ViT over CNNs on large datasets, it can be seen that ViT has its advantages. Studies [23] show that compared with CNNs, ViT has more similarities between the representations obtained in lower and higher layers and integrates more global information than ResNet in lower layers. Moreover, the skip connection in ViT is more influential than that in ResNet, which has a strong impact on performance and the similarity of representation.

This paper proposes an improved algorithm, ReVI, based on ViT to classify bamboo images. The core of ReVI takes ViT as the main framework and combines the convolutional and residual structure with multi-head attention modules. The bamboo image samples collected in Section 2.1 below are displayed. Section 2.2 below first shows the background of the ReVI algorithm improvement, the specific improvement method and the process; Section 3.1 shows the characteristics and preprocessing process of bamboo data; Section 3.2 focuses on comparing the performance of ReVI with ViT and tuning ReVI; and Section 3.3 compares ReVI with ViT and CNNs such as ResNet, Densenet, Xception, and then analyses the image classification results under different categories of bamboo. Finally, the reasons for ReVI's good performance were analyzed.

## 2. Experiment and Methods

### 2.1. Bamboo Resources

The images of bamboo resources in this paper were taken in Hengshan Forest Park, Guangde County, Xuancheng city, Anhui Province, which is a significant place for national bamboo forest germplasm resources. A Canon EOS 6D camera was employed, and the features included a Canon EF 24–105 mm, 26.2-megapixel complementary metal-oxide semiconductor (CMOS) sensor with dual-pixel CMOS auto focus (AF). To make the classification results more general, images of bamboo from different angles, such as the height of bamboo, thickness of bamboo poles and texture of bamboo leaves, were collected. The bamboo images were then labeled, after which they were collated into a bamboo dataset

with a total of 3220 images of 19 species. The number of images in each species varied. Figure 1 shows a sample of 19 species of bamboo.

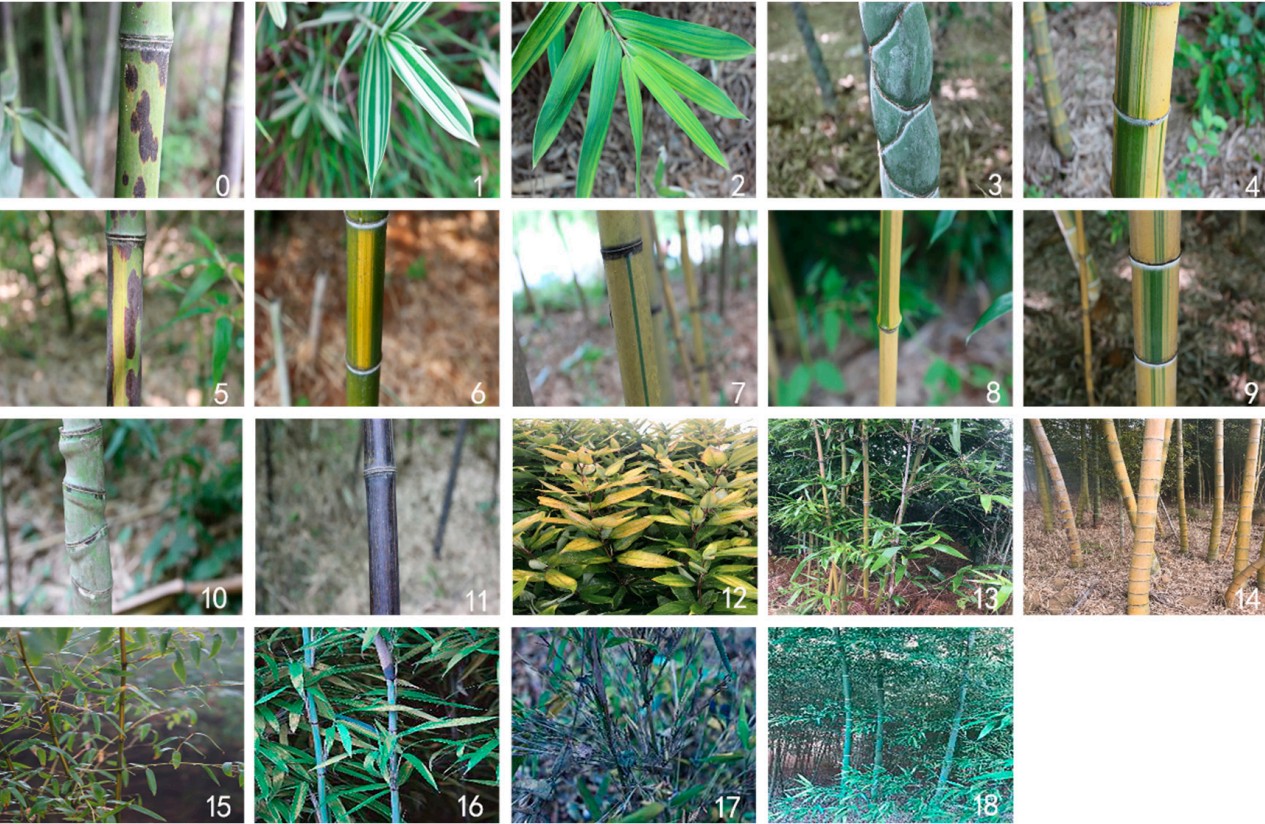

**Figure 1.** Nineteen species of bamboo sample [1,24]. Note: Nos. **0**–**4** are Phyllostachys bambusoides Sieb. et Zucc. f. lacrima-deae Keng f. et Wen, Pleioblastus fortunei (v. Houtte) Nakai, Pleioblastus viridistriatus (Regel) Makino, Phyllostachys edulis 'Heterocycla', Phyllostachys eduliscv. Tao kiang; Nos. **5**–**9** are Phyllostachys bambusoidesf.mixtaZ.P.WangetN. X. Ma, Phyllostachys heterocycla (Carr.) Mitford cv. Luteosulcata Wen, Phyllostachyssulphurea (Carr.) A. et C. Riv. 'Robert Young', Phyllostachys aureosulcata 'Spectabilis' C.D. Chu. Et C.S. Chao, Phyllostachysheterocycla (Carr.) Mitford cv. Viridisulcata; Nos. **10**–**14** are Phyllostachys aurea Carr. ex A. et C. Riv, Phyllostachys nigra (Lodd.) Munro, Shibataea chinensis Nakai, Acidosasa chienouensis (Wen.) C. C. Chao. et Wen, Bambusa subaequalis H. L. Fung et C. Y. Sia; Nos. **15**–**18** are Phyllostachys hirtivagina G.H. Lai f. flavovittata G.H. Lai, Pseudosasa amabiLis (McClure) Keng f, P.hindsii (Munro) C.D. Chu et C.S. Chao, Phyllostachys heterocycla (Carr.) Mitford cv. Obliquinoda Z.P. Wang et N.X. Ma.

*2.2. Methods*

2.2.1. Vision Transformer

CNNs continue to evolve and dominate in image classification tasks [25–28]. As CNNs develop, their depth also increases. It has been found that as the depth of a CNN increases, the problem of gradient explosion arises, and adding more layers to a saturated model leads to higher training errors. ResNet, which is based on a residual structure, solves this degradation problem of deep convolutional neural networks [11]. The residual structure of ResNet is shown in Figure 2 below.

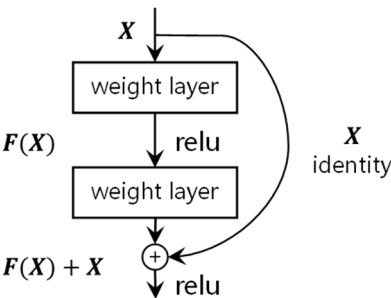

**Figure 2.** Residual block [11].

The desired base mapping is denoted as H(X), making the stacked nonlinear layers fit into another mapping of F(X): F(X) = H(X) − X. The original mapping is reconstructed as F(X) + X. This new mapping F(X) + X allows for a shotcut connection [29–31] of one or more layers of the neural network, and this mapping simply adds identity mapping directly to the output of the stacked layers, adding neither additional parameters nor computational complexity. A ResNet network sets up a residual network with a depth of up to 152 layers on the ImageNet classification dataset, 8 layers deeper than the VGG network, but still with lower complexity and achieving an error of 3.57%, making it the most-used CNN for recent image classification tasks.

In recent years, the transformer algorithm has improved the parallelization and computational efficiency of recurrent neural networks [32,33] (RNNs), which have been applied successfully to NLP tasks, such as language modeling and machine translation [34–36]. Therefore, some studies have applied transformers to image classification tasks and surpassed CNNs such as ResNet18 on large datasets, but the performance on small datasets is slightly worse than CNNs. Therefore, this paper cites the ViT algorithm, a variant model of a transformer for image classification tasks, to analyze its performance on small datasets through classification results on bamboo datasets and then improve ViT.

The core structure of a transformer network is the attention mechanism. An attention mechanism allows dependencies in a sequence of inputs or outputs to be modeled without regard to the distance between them. It has become an integral part of compelling sequence modeling and transformation models for a variety of tasks [35,37], and self-attention is an attention mechanism that calculates the information of a single sequence by associating different positions of the sequence [15]. Self-attention has been successfully applied to a wide variety of tasks, including reading comprehension [35], textual entailment [38], and abstract generalization [39]. An attention function is a mapping of a query and a set of key and value to an output, where query, key and value are vectors. The output of the attention mechanism is a weighted sum of values, where the weight assigned to each value is calculated by the compatibility function of the query with the corresponding key. Two variants of an self-attention mechanism are scaled dot-product attention and multi-head attention. The input to the scaled dot-product attention is the query matrix $Q$ of dimension $d_k$, the key matrix $K$ and the value matrix $V$ of dimension $d_v$. $Q$ and $V$ are dotted and divided by the root $d_k$, and then a softmax function is used to obtain the weights of $V$, which are multiplied by $V$. This is shown below.

$$\text{Attention}(Q, K, V) = \text{softmax}\left(\frac{QK^T}{\sqrt{d_k}}\right)V \tag{1}$$

The multi-head attention mechanism is a linear projection of query, key and value into multiple subspaces, each subspace being a $head_i$, which computes its scaled dot-product attention value separately, and these values connected and projected again to obtain the final value:

$$\text{MultiHead}(Q, K, V) = \text{Concat}(head_1 \ldots, head_h)W^O$$

$$\text{Where } head_i = \text{Attention}(QW_i^Q, KW_i^K, VW_i^V)$$

where the projections are parameter matrices $W_i^Q \in \mathbb{R}^{d_{model} \times d_k}$, $W_i^K \in \mathbb{R}^{d_{model} \times d_k}$, $W_i^V \in \mathbb{R}^{d_{model} \times d_v}$ and $W^O \in \mathbb{R}^{hd_v \times d_{model}}$.

A transformer is the first transduction model to rely entirely on a self-attentive mechanism to compute its input and output representations without the use of RNNs or convolution [15]. A transformer uses an encoder-decoder structure, as shown in Figure 3 below.

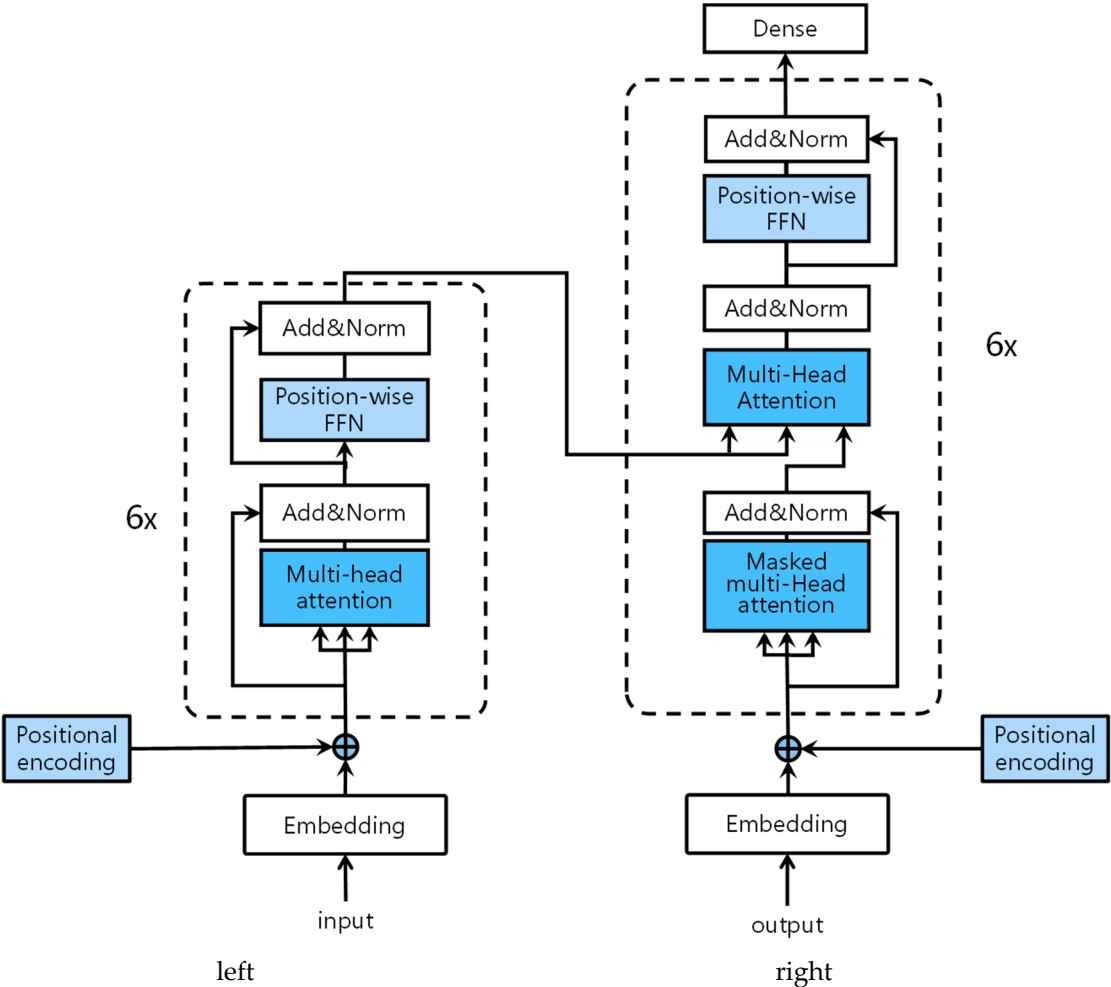

**Figure 3.** Transformer encoder-decoder architecture. The figure on the **left** is the encoder of the transformer and the figure on the **right** is the decoder of the transformer [15].

The left side of Figure 3 shows the encoder [15]. The right side of Figure 3 shows the decoder. The encoded embedding is added to the positional encoding and fed into the encoder and decoder. The encoder consists of six stacks, each with the same composition, with two sublayers. The first sublayer is a multi-head attention mechanism. The second sublayer, a position-wise FFN, is a simple, fully connected feedforward network. The residual connection is used to surround each sublayer, and then Add&Norm performs layer normalization [40]. The output of each sublayer is LayerNorm (x + Sublayer (x)), where Sublayer (x) is a function implemented by the sublayer itself. The decoder is also made up of six identical stacks. In addition to the two sublayers in each encoder layer, the decoder inserts a third identical sublayer, a multi-head attention mechanism, which performs the multi-head attention function on the output of the encoder stack. Similar to the encoder, each sublayer is connected using a residual connection around each sublayer and then through layer normalization. The sublayer multi-head attention in the decoder

stack is modified to masked multi-head attention to prevent the current position from noticing subsequent positions. This ensures that predictions for position *i* can only depend on known outputs for positions less than *i*. Finally, the result is output through a dense linear layer.

### 2.2.2. Residual Vision Transformer Algorithm

ViT [22] is a variant model of a transformer in the field of computer vision that uses the encoder of the transformer network with some adaptations to the original transformer encoder. Figure 4 shows the framework of ViT.

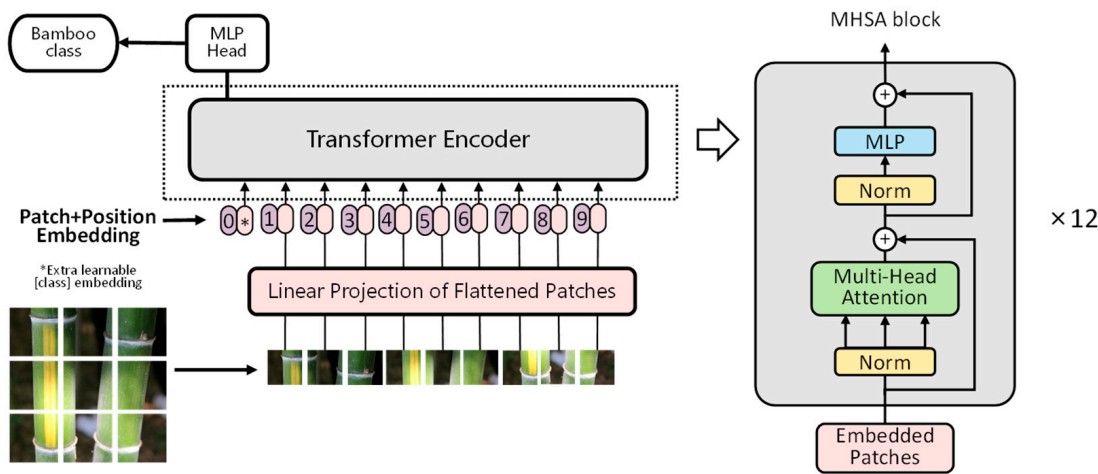

**Figure 4.** The framework of ViT [22].

The input to a standard transformer is a 1D sequence of embeddings. To process a 2D image, ViT cuts each image of height *H* and width *W* into patches of length and width *p*, which transforms image $x \in \mathbb{R}^{H \times W \times C}$ into a flattened sequence of 2D patches $x_p \in \mathbb{R}^{N \times (P^2 \cdot C)}$, where $(H \times W)$ is the resolution of the original image, *C* is the number of channels, $(P, P)$ is the resolution of each patch, and $N = HW/P^2$ is the number of patches and the input sequence length for ViT. ViT passes the flattened patches to the *D* dimension using a trainable linear projection. The output of this projection is patch embeddings. Similar to BERT's [class] token [19], we added a learnable embedding (named [class embedding] [22]) to the sequence of embedded patches. Then, standard learnable 1D position embeddings were used, which were added to the patch embeddings to preserve position information. The resulting sequence of embedding vectors was used as input to the transformer encoder. ViT's transformer encoder has 12 multi-head self-attention [22] (abbreviated MHSA) blocks. Each MHSA block consists of alternating layers of multi-headed self-attention and MLP, with layer normalization (Norm) applied before each MHSA block and residual connections [11] added before each Norm.

ViT features a pure transformer network with some transformations to suit the image classification task. ViT, based on multi-head attention, has been confirmed to outperform CNNs on large-scale datasets but slightly underperforms CNNs on small-scale datasets [22]. The poor performance of ViT models on small-scale datasets is mainly due to the difference in image information extraction between CNN and multi-head attention. A multi-head attention mechanism can extract the global information of an image, and large-scale data aggregate this global information earlier [23]. The convolutional structure has inductive bias such as locality, two-dimensional neighborhood structure and translation equivalence [22], and the current state-of-the-art CNN model, ResNet, adds identity mapping [11] to the original convolutional structure, making the CNN more complete. In contrast, in ViT,

only the MLP block incorporates residual connection [11,22]. The aim of this paper was to optimize and improve the architecture of a ViT model on small-scale datasets by embedding the convolutional layer and the residual unit of ResNet in the ViT model, which enables ViT to extract the inductive bias information of images in small-scale datasets, thus designing the improved algorithm ReVI.

　　The framework of the ReVI algorithm is shown in Figure 5. First, a convolutional layer and a residual unit were added before the first layer of the ViT model. The output of the residual unit was a feature map of the images after convolutional feature extraction. Then, the embedding layer of the ViT model was reconstructed, which is the image encoding layer, named the re_embedding layer. Unlike the embedding layer of the original ViT, which directly segments an image into patches, the re_embedding layer segments the output feature map of the residual unit into patches. The input image resolution is (368, 368), and the embedding layer of the original ViT directly segments the image into patches. The size of a patch is 16, and the number of patches is $23 \times 23$. The resolution of the feature map output from the residual unit of ReVI is (48, 48). To extract the association information between the features to a greater extent in the feature map, this paper changed the patch size to 2; the resolution of each patch is (2, 2), and the number of patches is $24 \times 24$. In this paper, the ReVI algorithm was tuned to find the optimal ReVI model for the bamboo dataset. The tuning step changes the size of the transformer encoder, which is the encoder of ViT. As shown in Figure 4, the original transformer encoder of ViT has 12 MHSA blocks. This paper reconstructed the transformer encoder and the number of MHSA blocks X as the transformer encoder parameters, adjusting X to construct ReVI_Xb $\in$ {ReVI_1b, ReVI _2b, ReVI _3b, . . . ReVI _12b, X $\in$ [1, 2, 3 . . . , 12]}. The network structure of ReVI is shown in Table 1 below.

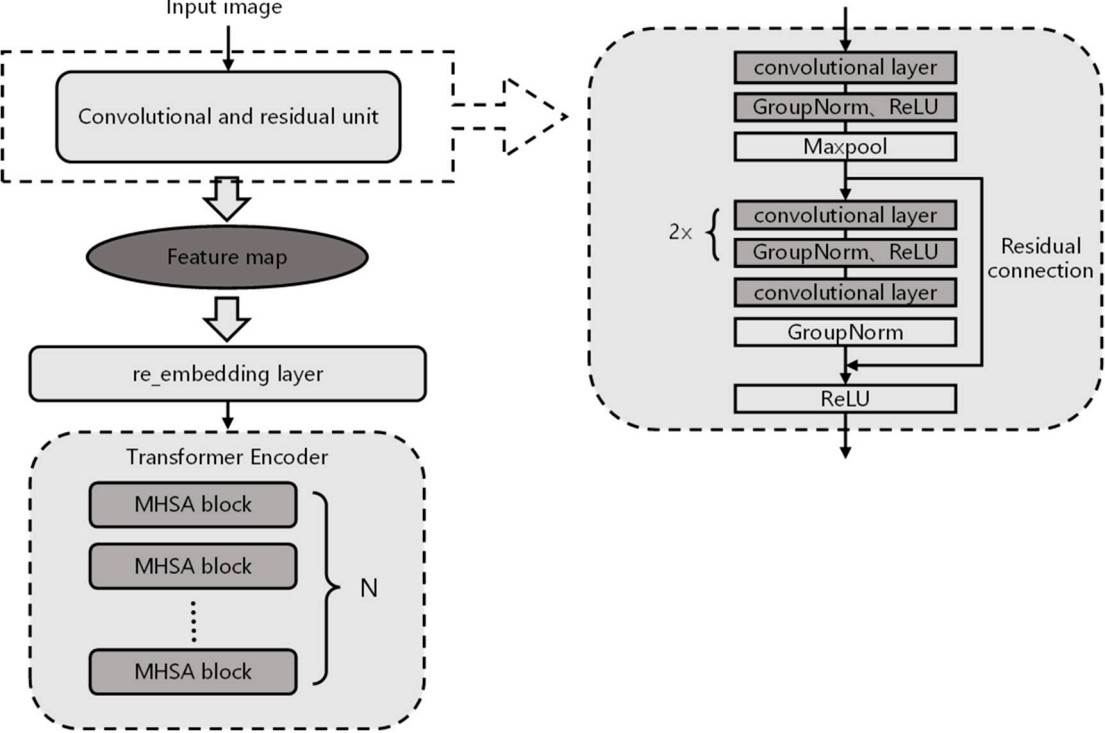

**Figure 5.** The framework of ReVI based on ViT [22], Transformer [15] and ResNet [11].

**Table 1.** ReVI main network layers and parameters.

| Layer | Output Size | Parameter |
|---|---|---|
| Conv2d | $192 \times 192$ | $7 \times 7$, 64, stride 2, padding 3 |
| Maxpool | $95 \times 95$ | $3 \times 3$, stride 2 |
| Residual unit | $48 \times 48$ | $\begin{pmatrix} 1 \times 1, & 128 \\ 3 \times 3, & 128 \\ 1 \times 1, & 256 \end{pmatrix} \times 1$ |
| Re_embedding layer | $768 \times 577$ | Patch_size = 2 |
| Transformer encoder | 768 | — |
| FC | 19 | — |

The kernel size of the ReVI convolutional layer is set to 7, the stride size is 2, the padding size is 3, the image output size is $192 \times 192$ after the convolutional layer, and the output size is $95 \times 95$ after the pooling layer. The residual unit includes three convolutional layers, with convolutional kernels of sizes 1, 3 and 1, and the residual unit also includes group normalization (GroupNorm) [41], ReLU activation function and the maximum pooling layer (Maxpool: kernel size 3, stride size 2) after the residual block to obtain a feature map of size $48 \times 48$. In this paper, we adjusted the original embedding_layer, and the modified encoding layer is the re_embedding layer. We changed the patch size of the re_embedding layer from 16 to 2. The number of patches is $24 \times 24 = 576$, which is 577 after adding class_token. The output size is $768 \times 577$ after passing through the encoder transformer encoder, and finally, the class probability is output through the fully connected layer FC.

ReVI combines the convolution and residual modules of CNNs with the ViT algorithm. Compared to ViT, which directly segments and encodes an image, ReVI segments and encodes the feature map output from the residual unit, reconstructs the embedding layer and ViT's encoding layer, and changes the size of the patch. The parameters of the encoder transformer encoder of ReVI were then adjusted by changing the number of MHSA blocks to select the optimal ReVI model. We compared the ReVI model with ViT, ResNet18 and VGG16 to explore the advantages of ReVI and compared ReVI with ViT by changing the proportion of the training dataset.

2.2.3. Quantitative Evaluation Indicators

A confusion matrix is an $n \times n$ table used to record the predictions of the n-element classifier. True positive (TP) is the number of positive classes predicted as positive. True negative (TN) is the number of negative classes predicted as negative. False positive (FP) is the number of negative classes predicted as positive. False negative (FN) is the number of positive classes predicted as negative classes. Accuracy (Accuracy, Acc, %), which is the number of samples correctly classified divided by the total number of samples, is calculated using the following formula:

$$\text{Acc} = \frac{\text{TP} + \text{TN}}{\text{TP} + \text{TN} + \text{FP} + \text{FN}} \times 100\% \tag{2}$$

Precision (precision, P, %), which is the proportion of the number of samples predicted to be in the positive category that is actually in the positive category, is calculated as follows:

$$\text{P} = \frac{\text{TP}}{\text{TP} + \text{FP}} \times 100\% \tag{3}$$

Recall (recall, R, %), which is the proportion of the sample predicted to be a positive class that is actually a positive class, is calculated as follows:

$$\text{R} = \frac{\text{TP}}{\text{TP} + \text{FN}} \times 100\% \tag{4}$$

The F1-score (F1, %) is a combination of precision and recall. The P and R sometimes contradict each other. The F1 combines the results of P and R. When the F1 is high, it can show that the classification method is effective. The calculation method is as follows:

$$F1 = \frac{2 \times P \times R}{P + R} \times 100\%$$ (5)

Specificity (false positive rate, FPR, %) is the proportion of negative samples misclassified as positive to all negative samples and is given by the following formula:

$$FPR = \frac{FP}{FP + TN} \times 100\%$$ (6)

The ROC curve (receiver operating characteristic) and the PR curve (precision–recall curve) are metrics for evaluating classifiers. The area under the ROC curve is the AUC, and the closer the AUC is to 1, the better the classification performance of the model. The AP (average precision) is the area under the PR curve, and the better the classifier, the higher the AP value. The mAP (mean AP) is the mean of the AP of multiple categories. mAP must be in the range [0, 1]; the larger, the better.

## 3. Results and Discussion

### 3.1. Bamboo Resource Images Dataset Analysis

In this paper, a total of 3220 images of 19 species of bamboo resources were collected. Figure 6 shows the statistics of the number of bamboo images in each class, which were divided into a training dataset, validation dataset and test dataset at a ratio of 7:2:1. Due to the imbalance in the number of samples in each category, some images are augmented by flipping, cropping, adding pretzel noise, etc.

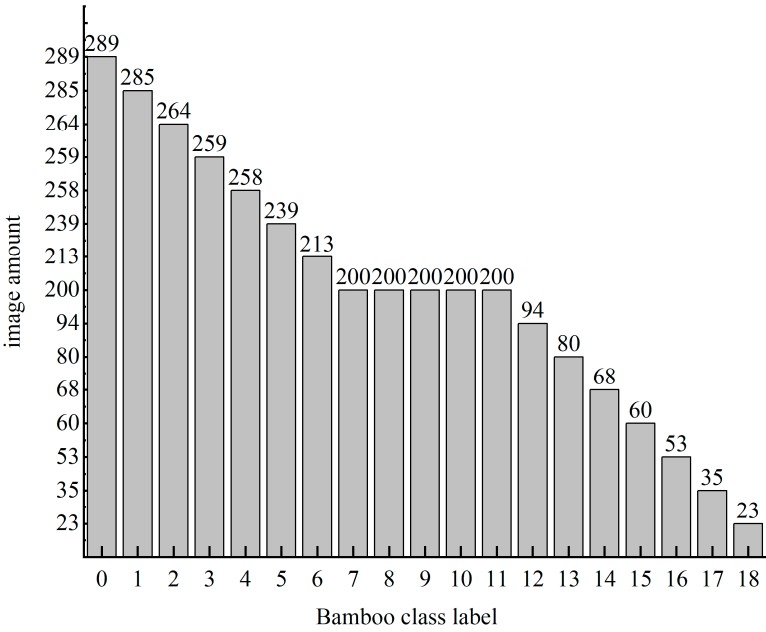

**Figure 6.** Quantity statistics of each class of the bamboo dataset.

The software environment used in this paper is Ubuntu, the open-source PyTorch framework and Python programming language (Python 3.8). The hardware is a computer with 16 GB of RAM, Intel(R) Core i7-9700k, a 3.60 GHz processor and an NVIDIA GeForce RTX 1080Ti GPU with 8 GB of video memory. The learning rate during model training and validation was 0.001, the momentum factor was 0.9, the number of training epochs was 100, and the number of batches was 8. A transform function was used to preprocess an image before input to the model, which randomly cropped images to a 384 × 384 size and flipped

and transformed images into tensors, and tensors were normalized using mean= [0.485, 0.456, 0.406] and standard deviation (std) = [0.229, 0.224, 0.225] as the input to the model.

To explore the performance of ReVI, none of the models in this paper used fine tuning; instead, the network parameters were self-initialized. All convolutional layer weight parameters were initialized using a Kaiming normal distribution, which obeys $\mathcal{N}\left(0, \text{std}^2\right)$. The std is calculated as follows:

$$\text{std} = \frac{\text{gain}}{\sqrt{\text{fan\_mode}}} \tag{7}$$

The gain is a proportional value to regulate the relationship between the input order of magnitude and the output order of magnitude, which is determined by the activation function. The fan_mode is determined by the number of weight parameters and the direction of propagation. For the fully connected layer, fan_in is the input dimension, and fan_out is the output dimension. For the convolutional layer, its dimension was set to $[C_{out}, C_{in}, H, W]$, where $H \times W$ is the kernel size, $C_{out}$ is the number of output channels and $C_{in}$ is the number of input channels. The convolution layer bias is initialized to 0. All linear layer weight parameters were initialized using a Kaiming normal distribution, which follows $\mathcal{N}\left(0, \text{std}^2\right)$. The std is calculated as follows:

$$\text{std} = \text{gain} \times \sqrt{\frac{2}{\text{fan\_in} + \text{fan\_out}}} \tag{8}$$

The position embeddings of ViT were initialized to a normal distribution $\mathcal{N}(0, 1)$ and [class] embedding was initialized to 0.

### 3.2. Analyzing the Performance of ReVI on Bamboo Datasets

The main structure of the ReVI algorithm used in this paper is ViT. The model scale of the ViT algorithm is too large on the bamboo dataset, and the overfitting phenomenon is serious. In this paper, the model size of ReVI is adjusted without changing the model architecture. The parameters of the transformer encoder of ReVI are adjusted by changing the number of MHSA blocks, and then ReVI_Xb $\in$ {ReVI_1b, ReVI _2b, ReVI _3b, ... ReVI _12b, $X \in [1, 2, 3 ..., 12]$} is constructed. When $X \in [6, 7, 8, 9, 10, 11, 12]$, ReVI_Xb is discarded because it is too large and overfitted. The ReVI_Xb(X∈[1,2,3,4,5]) model was trained for validation and testing, and Table 2 shows the highest accuracy ReVI_Xb($X \in [1, 2, 3, 4, 5]$) model on the training, validation and test datasets.

**Table 2.** ReVI tuning model accuracy.

| Highest Accuracy/% | ReVI_Xb(X∈[1,2,3,4,5]) | | | | |
| --- | --- | --- | --- | --- | --- |
| | ReVI_1b | ReVI_2b | ReVI_3b | ReVI_4b | ReVI_5b |
| Training dataset | 95.15 | 96.18 | 96.18 | 96.27 | 96.58 |
| Validation dataset | 96.65 | 95.34 | 95.81 | 95.50 | 95.65 |
| Test dataset | 87.46 | 89.30 | 90.21 | 88.07 | 89.60 |

As seen in Table 2, the model training validation and test accuracy of ReVI_Xb are relatively high, and the model generalization ability is better than that of the other models. The ReVI test accuracy is highest at X = 3 and starts to decline when $X \in [3, 4, 5]$. The accuracy of the combined training and test dataset shows that model overfitting occurs when $X \in [3, 4, 5]$. In this paper, for ReVI, the current study further visualizes the accuracy change trend on the test dataset, as shown on the left side of Figure 7. Then, the current work analyzed the accuracy, recall and F1-score for each class of bamboo images, taking and visualizing the accuracy, average recall and average F1-score, average specificity and

average mAP for the 19 bamboo species on the test dataset, as shown on the right side of Figure 7.

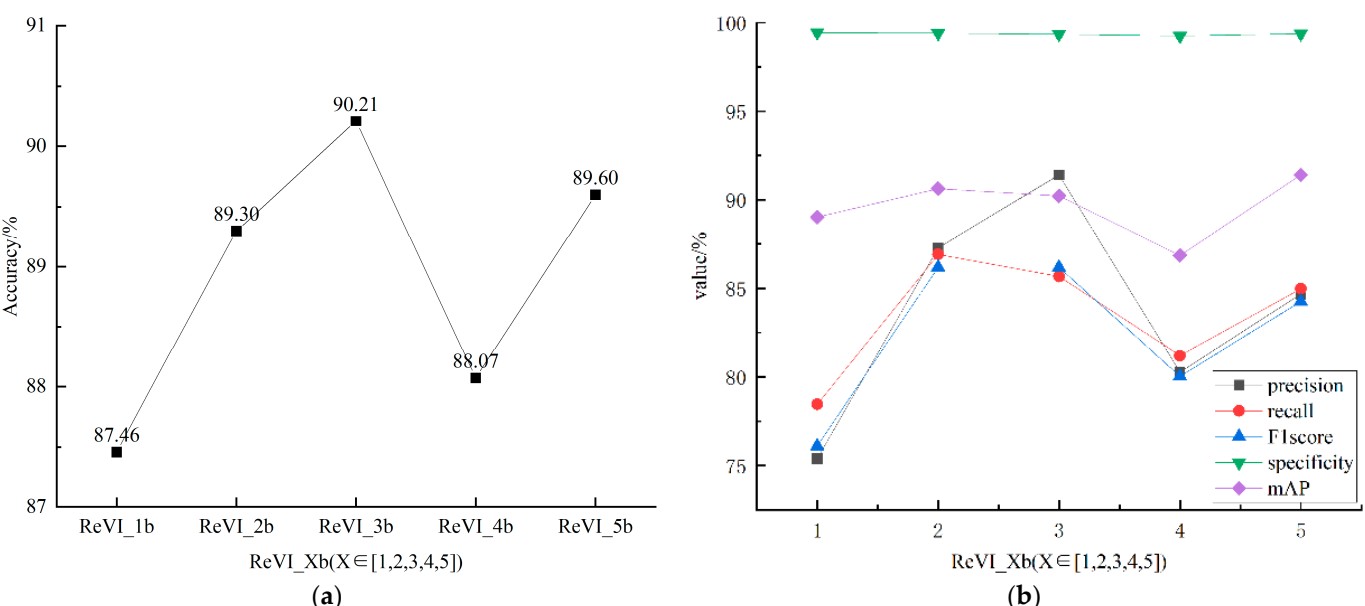

(**a**)              (**b**)

**Figure 7.** The performance of the ReVI models: (**a**) accuracy variation curve of the ReVI_Xb$(X \in [1, 2, 3, 4, 5])$; (**b**) average accuracy, average recall and average F1-score and average specificity variation curves of the ReVI_Xb$(X \in [1, 2, 3, 4, 5])$.

As seen in Figure 7a, the generalization ability of the ReVI_Xb model gradually increases when $X \in [1, 2, 3]$, with the highest accuracy being 90.21%, and the generalization ability starts to decrease when $X \in [3, 4, 5]$. As seen in the Figure 7b, the ReVI_Xb model shows an increasing trend in average accuracy, average recall and average F1-score when $X \in [1, 2, 3]$ and a decrease at $X \in [3, 4, 5]$. It can be concluded that the ReVI_3b model is the best ReVI model.

To explore the effect of the convolution and residual structure on the ViT algorithm, ViT was optimized, and ViT_Xb $\in \{$ViT_1b, ViT _2b, . . . ViT _5b, $X \in [1, 2, 3, 4, 5]\}$ were constructed and compared with ReVI_Xb $\in \{$ReVI_1b, ReVI _2b, . . . ReVI _5b, $X \in [1, 2, 3, 4, 5]\}$. Figure 8 shows the training loss and accuracy variation curves.

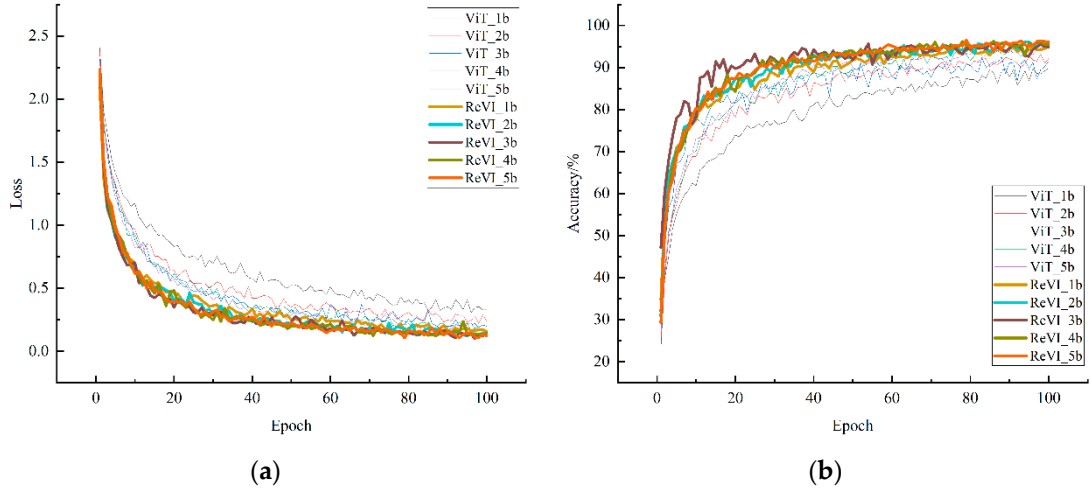

(**a**)              (**b**)

**Figure 8.** (**a**) Training loss variation curve of the ViT_Xb$(X \in [1, 2, 3, 4, 5])$ and ReVI_Xb$(X \in [1, 2, 3, 4, 5])$; (**b**) accuracy variation curve of the ViT_Xb$(X \in [1, 2, 3, 4, 5])$ and ReVI_Xb$(X \in [1, 2, 3, 4, 5])$..

The thin dashed line in Figure 8 shows the variation curve of ViT_Xb($X \in [1, 2, 3, 4, 5]$), and the thick dashed line shows the variation curve of ReVI_Xb($X \in [1, 2, 3, 4, 5]$). Figure 8 shows that the convergence rate of the ReVI_Xb($X \in [1, 2, 3, 4, 5]$) model is faster than that of ViT_Xb($X \in [1, 2, 3, 4, 5]$), and the prediction accuracy of the model after convergence is higher than that of ViT_Xb($X \in [1, 2, 3, 4, 5]$). The current study then tested the ReVI_Xb($X \in [1, 2, 3, 4, 5]$) and ViT_Xb($X \in [1, 2, 3, 4, 5]$) models constructed after training and validation to test the generalization ability of both on the test dataset. Figure 9 shows the prediction accuracy of both on the test dataset.

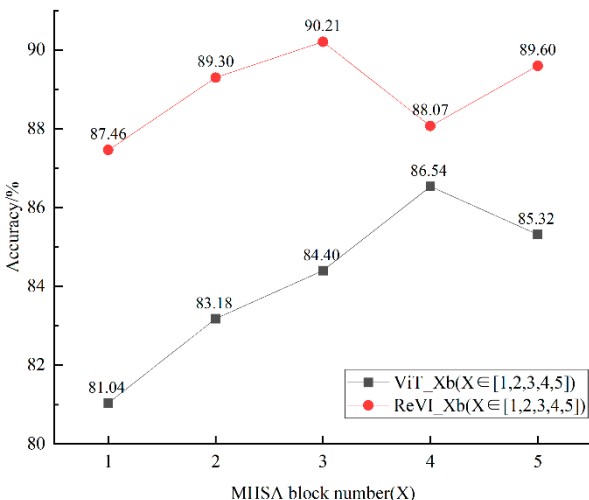

**Figure 9.** Prediction accuracy of ViT_Xb($X \in [1, 2, 3, 4, 5]$) and ReVI_Xb($X \in [1, 2, 3, 4, 5]$) on the test dataset.

As shown in Figure 9, the prediction accuracy of ReVI_Xb($X \in [1, 2, 3, 4, 5]$) is higher than that of ViT_Xb($X \in [1, 2, 3, 4, 5]$), indicating that the generalization ability of ReVI is better than that of ViT. Then, by analyzing the respective tuning results of ReVI_Xb($X \in [1, 2, 3, 4, 5]$) and ViT_Xb($X \in [1, 2, 3, 4, 5]$), the highest accuracy of ReVI_Xb($X \in [1, 2, 3, 4, 5]$) is 90.21%, which is the optimal ReVI model. Similarly, it can be concluded that the optimal ViT model is ViT_4b, with an accuracy of 86.54%.

The precision, recall and F1-score of the models on the 19 classes of bamboo data were analyzed and compared. Table 3 shows the average accuracy, average recall, average F1-score, average specificity and average mAP of ReVI_3b and ViT_4b. As shown in Table 3, the average accuracy, average recall, average F1-score, average specificity and average mAP of ReVI_3b are mostly higher than those of ViT_4b.

**Table 3.** Average precision, recall, F1-score, specificity and mAP of ViT_4b and ReVI_3b.

| Model | Precision/% | Recall/% | F1-Score/% | Specificity/% | mAP/% |
|---------|-------------|----------|------------|---------------|--------|
| ReVI_3b | 91.40 | 85.67 | 79.63 | 99.35 | 91.00 |
| ViT_4b | 82.89 | 83.39 | 80.77 | 99.19 | 90.22 |

From the above results, it can be concluded that the ReVI algorithm outperforms the ViT algorithm on bamboo data. The convolutional layers and residual blocks enable ViT to improve the convergence speed and accuracy of model training and improve the generalization ability with a 3.67% increase in accuracy, 8.51% increase in average precision, 2.28% increase in recall, 0.16% increase in specificity and 0.78% increase in mAP on bamboo data

It has been shown [22] that when training on mid-scale and small-scale datasets, ViT has poorer performance than CNNs due to the lack of inductive bias information in CNNs. Therefore, the effect of training sample size on ViT is explored by adjusting the size of the training dataset, and whether the ReVI enables ViT to overcome this deficiency is

explored. We originally divided the bamboo dataset with a ratio of 7:2:1. We then reduced the proportion of the training dataset to 5 and 3, which changed the division ratio to 5:2:1 and 3:2:1. ReVI_3b and ViT_4b were used to train, validate and test on the datasets with the three division ratios; the performance of ReVI_3b and ViT_4b were compared, and the accuracy on the test dataset is shown in Figure 10.

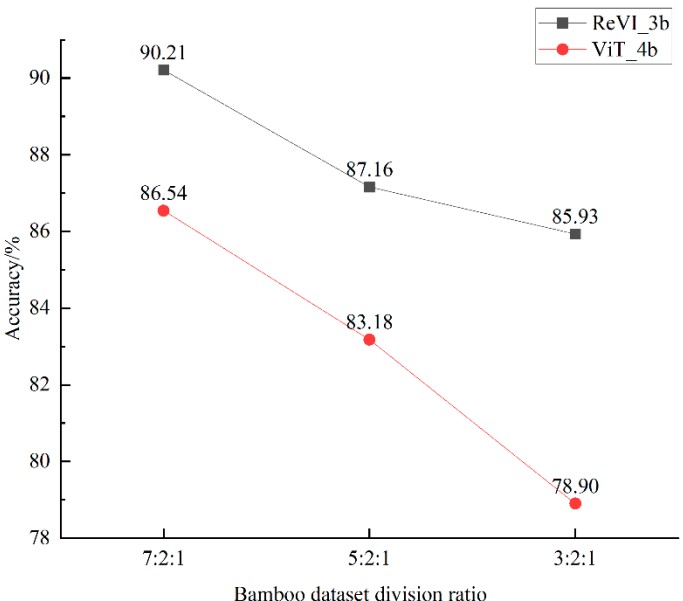

**Figure 10.** Accuracy of ReVI_3b and ViT_4b on the test dataset with three division ratios.

The accuracy of ReVI_3b is higher than that of ViT_4b for all three dataset division ratios, and the accuracy of ReVI_3b decreases less than that of ViT_4b. The accuracy of ReVI_3b changes more steadily as the proportion of the training dataset decreases, while that of ViT_4b changes more sharply. It can be concluded that changes to the amount of training dataset cause a significant impact on ViT, but not ReVI, so it can be concluded that ReVI overcomes the impact of the training amount on the performance of the ViT algorithm.

After a series of tuning and comparisons, it was concluded that ReVI_3b was the optimal model, and a preliminary analysis of the class performance of ReVI_3b for the model was carried out.

As can be seen in Figure 11a,b, ReVI_3b has a good ROC curve and PR curve on the test set, with an AUC close to 1.0 for all categories and a better AP for each category, with only individual classes performing somewhat poorly.

### 3.3. Comparison with Different Deep Learning Models

In order to investigate whether ReVI outperforms CNN and ViT [22], this paper compared the performance of ReVI_3b, ViT_4b, ResNet18 [11], ResNet50 [13], VGG16 [9], DenseNet121 [12] and Xception [42]. To explore the performance of the above models on the bamboo data, this paper initialized the network parameters of ViT_4b and ResNet18, ResNet50, VGG16, DenseNet121 and Xception so that they are trained entirely on the bamboo dataset. It is well known that fine-tuning is a popular method for transferring large models to small-scale datasets [16,18], so in this paper, we also included the pretraining ResNet18 (pretraining) and ResNet50 (pretraining); all models were trained for 100 epochs, other hyperparameters such as batch_size were the same, and the accuracy and loss variation of the training process was as Figure 12.

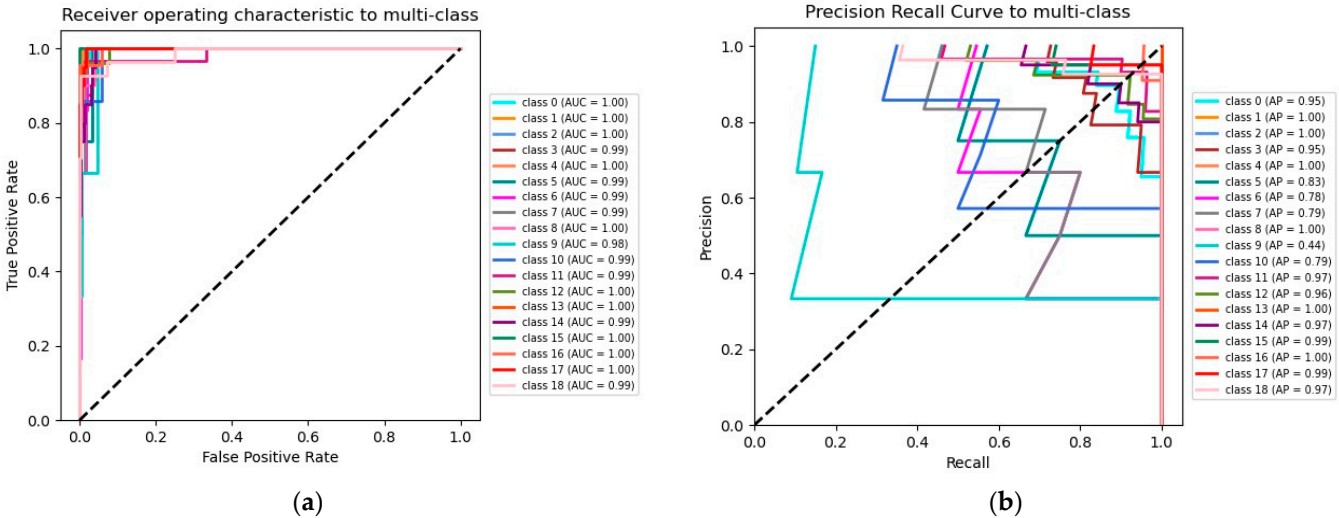

**Figure 11.** (**a**) ROC (receiver operating characteristic) curve of the ReVI_3b. (**b**) PR curve (precision–recall curve) of the ReVI_3b.

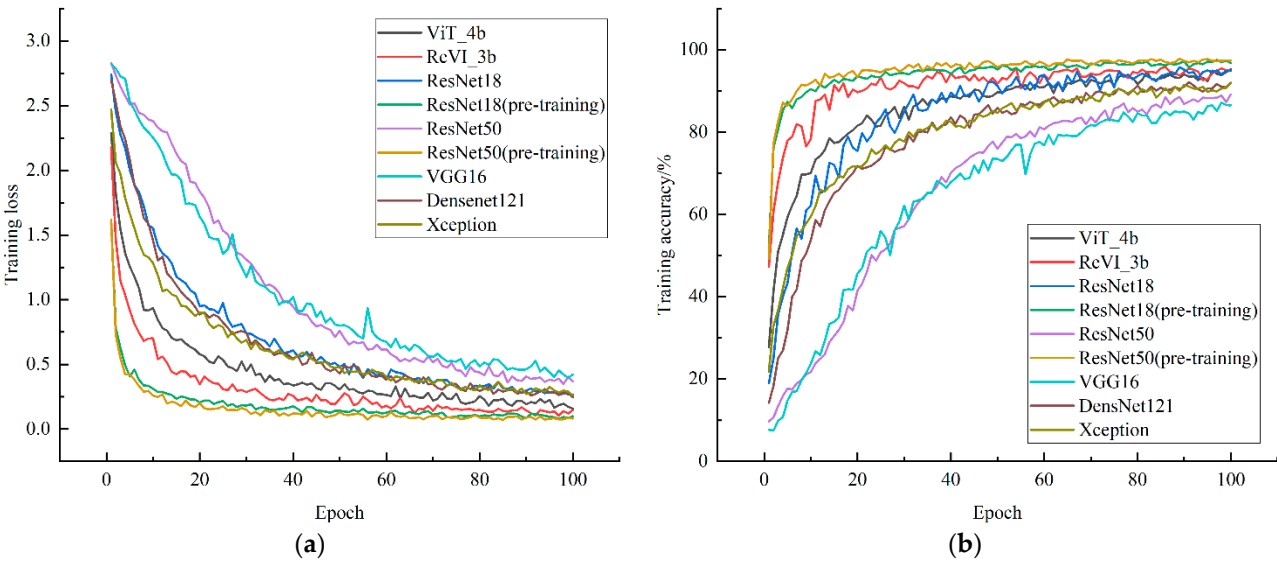

**Figure 12.** (**a**) Variation curves of loss for different deep models on the training dataset; (**b**) variation curves of accuracy for different deep models on the training dataset.

We can see that the loss and accuracy iterations of the ReVI model are faster than the other non-pretraining models, including ViT_4b, ResNet18, ResNet50, VGG16, DenseNet121 and Xception, and the convergence loss values and accuracy of the ReVI model are better than the above non-pretraining models. However, the performance of the training process of ReVI is relatively poor compared with the pretraining models ResNet18 and ResNet50, but the performance of the training process does not completely explain the problem, so we used the above models for the prediction of the test set, and the following Table 4 show the accuracy, average precision, average recall, average F1-score, average specificity and mAP on the test set.

As can be seen from Table 4, ReVI_3b outperformed ViT and all other non-pre-trained CNNs on the test set in terms of accuracy, average recall, average F1-score, average specificity and mAP. Whereas the high accuracy of ResNet18 (pretraining) and ResNet50 (pretraining) indicates that our dataset is fine, the accuracy of ReVI reached 90%, which is somewhat better than ResNet18, indicating that ReVI outperforms ViT and most CNNs on bamboo data.

**Table 4.** Predicted results of ReVI_3b and other models on test set.

| Model | ReVI_3b | ViT_4b | ResNet18 | ResNet50 | VGG16 | DenseNet121 | Xception | ResNet18 (Pretraining) | ResNet50 (Pretraining) |
|---|---|---|---|---|---|---|---|---|---|
| Training time | 60 m 20 s | 59 m 9 s | 40 m 34 s | 69 m 55 s | 88 m 29 s | 102 m 36 s | 74 m 10 s | 44 m 52 s | 62 m 50 s |
| accuracy/% | 90.21 | 85.63 | 84.71 | 87.16 | 84.4 | 85.93 | 82.97 | 88.07 | 94.5 |
| Precision/% | 80.15 | 81.55 | 85.44 | 82.08 | 83.52 | 81.68 | 75.19 | 80.23 | 93.9 |
| Recall/% | 80.83 | 81.26 | 75.56 | 82.73 | 75.81 | 77.38 | 76.12 | 81.02 | 87.38 |
| F1-score/% | 79.63 | 80.77 | 77.47 | 81.23 | 75.98 | 78.43 | 74.89 | 79.88 | 89.03 |
| Specificity/% | 99.35 | 99.19 | 99.14 | 99.29 | 99.12 | 99.21 | 99.04 | 99.34 | 99.69 |
| mAP/% | 90.22 | 91 | 89.94 | 88.73 | 83.99 | 88.9 | 86.86 | 85.86 | 96.36 |

The classification performance of ReVI_3b with ViT_4b, ResNet18 and VGG16 for each class of bamboo images was analyzed, and Figure 13 shows the accuracy of the models for each class in the test dataset.

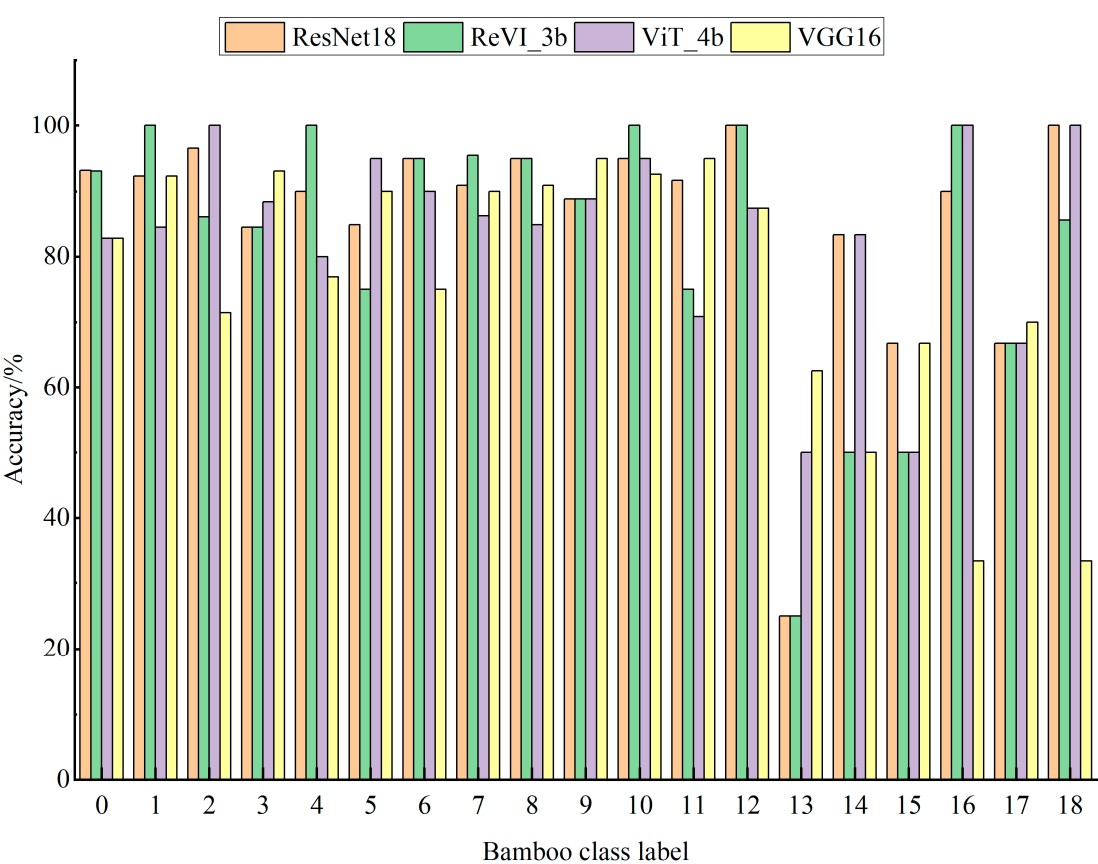

**Figure 13.** Classification results of different models for each class of bamboo.

Figure 13 shows that the classification accuracy of most bamboo species by each model is high, but the accuracy of classes 13, 14, 15, and 17 is low. We investigated the reasons for the low accuracy of these four classes using a confusion matrix on the test dataset, and the confusion matrix of ReVI on the test dataset is shown in Figure 13.

As seen in Figure 14, Class 13 has six images in total, two images misclassified as Class 16 and one image misclassified as Class 0. Class 14 has six images in total, with one image misclassified as Class 18 and two images misclassified as Class 3. Class 15 has six images in total, with two images misclassified as Class 17 and one image misclassified as Class 12. Class 17 has six images in total, with one image misclassified as Class 13. We analyzed the reasons for the poor classification of these four classes: 1. The number of samples is too few, resulting in the model not being able to learn enough features and the model generalization

ability being poor. 2. The features of classes 13, 14, 15, and 17 are not obvious enough and are easily confused with other classes of bamboo.

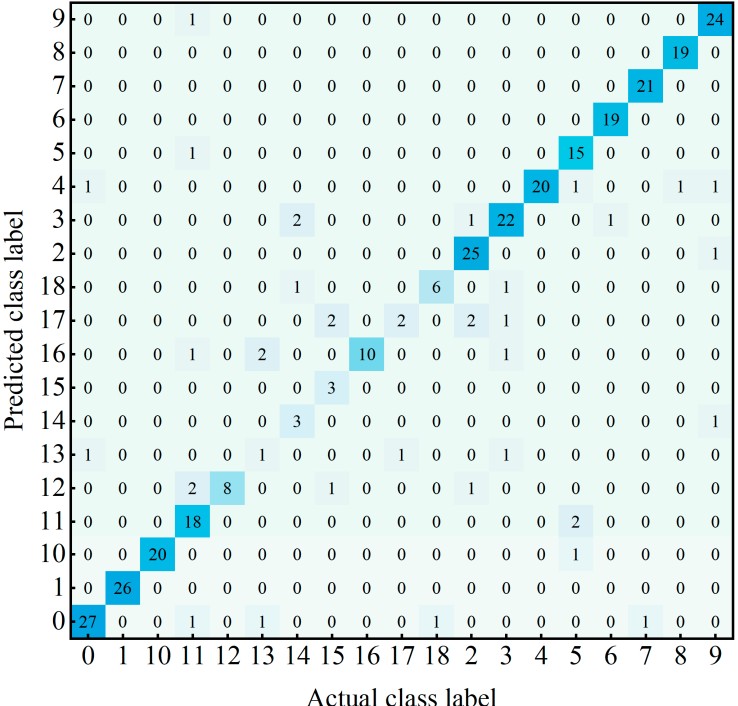

**Figure 14.** Confusion matrix of ReVI on the test dataset.

This section shows the classification performance of ReVI, ViT and CNNs such as ResNet and VGG16 on the bamboo dataset and then analyzes different evaluation metrics. From the above results, it can be concluded that the bamboo images can be classified and identified using deep neural networks such as CNNs and transform networks, both of which obtain high accuracy. The improved ViT algorithm is superior to the classical CNNs on the bamboo dataset, as evidenced by the fact that ReVI outperforms most CNNs in all evaluation metrics on the bamboo dataset. ReVI has a multi-head attention mechanism as its core architecture, and it can be seen that the attention mechanism performs well in image classification tasks, which is very important for image classification on complex backgrounds.

## 4. Conclusions

The image acquisition of bamboo species is difficult. Therefore, a better generalization of the deep learning algorithm is important for identifying different bamboo species. The ReVI algorithm improved the ViT algorithm by combining a convolutional residual mechanism with the multi-head attention mechanism on a dataset of bamboo species images. By comparing the performances of ReVI, ViT, ResNet and VGG16 on a bamboo dataset, it was concluded that ReVI outperformed ViT and CNNs. It has been demonstrated that ViT lacks inductive bias such as locality, two-dimensional neighborhood structure, and translation equivariance on small-scale datasets, which can be extracted using CNNs, while pretraining on large amounts of data allows ViT to outperform a CNN's inductive bias [22]. The residual mechanism proposed by He et al. in 2016 solves the network degradation caused by increasing the depth of CNNs and makes the convolutional structure more perfect [11].

The main contributions of this paper are:

1.  This paper improves the ViT algorithm, the ReVI algorithm proposed in this paper outperforms both the ViT and CNN algorithms on the bamboo dataset, and ReVI still outperforms ViT despite decreasing the number of training samples. It can be

concluded that the convolution and residual mechanisms compensate for the inductive bias that cannot be learned by ViT on a small-scale dataset, rendering ViT no longer limited to the number of training samples and equally applicable to classification on small-scale datasets.

2. Bamboo varies in species and is widely distributed around the world. Additionally, collecting bamboo samples requires much expertise and human resources. The average classification accuracy of ReVI is up to 90.21% compared to CNNs such as ResNet18, VGG16, Xception and Densenet121. The ReVI algorithm proposed in this manuscript can help bamboo experts to conduct more efficient and accurate bamboo classification and identification, which is important for the conservation of bamboo germplasm diversity.

Due to human and material constraints, the team was only able to collect a limited number of images. In future research, the team will collect more different types of data, continue to improve our algorithms, and deploy the models and data to servers to help classify bamboo images and use them for more image classification applications.

**Author Contributions:** Conceptualization, X.J.; methodology, Q.Z.; software, Y.S.; resources, L.W.; supervision, S.L., Y.R., X.Z. and Q.G.; funding acquisition, S.L., Y.R., X.Z. and Q.G. All authors have read and agreed to the published version of the manuscript.

**Funding:** This research was funded by the Natural Science Foundation of Anhui Province, grant number 2008085MF203; Anhui Provincial Natural Science Project, grant number KJ2019Z20 and KJ2019A0212; Anhui Province Key Research and Development Program Project, grant number 1804a07020108 and 201904a06020056.

**Data Availability Statement:** The data used in the manuscript are private data collected by the author and are not publicly available.

**Conflicts of Interest:** The authors declare no conflict of interest. The funders had no role in the design of the study.

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
