# Peer review of "An Improved Vision Transformer Network with a Residual Convolution Block for Bamboo Resource Image Identification"

_electronics, doi:10.3390/electronics12041055_

Round 1

Reviewer 1 Report

The reviewed paper: ”An improved vision transformer network with a residual convolution block for bamboo resource image identification”, introduce quite interesting new feasibilities of improved ViT algorithm. The authors have prepared the necessary photo sets for training, validate and testing analyzed artificial neural network algorithms. Based on assumed quality indices, researchers show that new proposed ReVI method, which combine the ViT, convolutional layer and the residual unit of ResNet, outperforms other classical methods (even Convolutional Neural Networks algorithm). The proposed approach seems to be useful not only in the context of classification of bamboo species.

Although the new approach is useful, some comments/issues arose during the review process. Below is list of mentioned drawbacks:

- line 17 – there is no space in: „work(ViT).”,

- line 60 – there is no space in: „BERT[16]uses”,

- line 86 – it would be good to add a brief note informing what will be in each paper section,

- line 98 – missing the statement: „[source: authors]”, if correct, or missing the reference,

- line 120 – missing the statement: „[source: authors]”, if correct, or missing the reference,

- the Figure 2. should be centralized,

- line 152 – is dot instead of comma: ” dimension dv. Q and V”,

- line 159 – missing final stop at the end of sentence,

- line 166 – missing the statement: „[source: authors]”, if correct, or missing the reference,

- line 181 – in the statement: ”This ensures that predictions for position i can only depend on known outputs for positions less than i.”,  the „i” should be italicized,

- all variables should be italicized, or if they are matrices or vectors, they should be bolded - please unify within whole paper,

- line 189 – missing the statement: „[source: authors]”, if correct, or missing the reference,

- line 221 – missing the statement: „[source: authors]”, if correct, or missing the reference,

- line 269 – there is a final stop at the end of sentence, but it should be after formula (2),

- line 272 – there is a final stop at the end of sentence, but it should be after formula (3),

- line 274 – there is a final stop at the end of sentence, but it should be after formula (4),

- line 277 – there is a final stop at the end of sentence, but it should be after formula (5),

- after formula (6) it should be a final stop,

- after formula (7) it should be a final stop,

- line 345 – there is no space between: ”.(b)”.

- line 369-372 – The presented data are true only for the special case considered in the reviewed paper. Therefore, it is necessary to refer the obtained results to the performed test, or to carry out further research in the context of the statistics of the repeatability of the obtained results.

- line 395 – there is no space between: ”.(b)”

- in Figure 11. there is final stop at the end of the caption, before there were no final stops at the end of other figures caption - please unify within whole paper

- line 397-399 – the sentence is unnecessary, in the Figure 11 there is a legend,

- line 452 – „The improved ViT algorithm is superior to classical CNNs for image classification tasks, as seen from the fact that all evaluation metrics of ReVI are optimal.”  - is optimal, but only for the analyzed test set - this should be emphasized.

After taking into account the above minor remarks, I believe that the work is suitable for publication in Electronics.

Reviewer 2 Report

The main contribution of this article should be stated via bullet after which follows the structure of the paper for smooth flow and understanding of the manuscript as well as good manuscript formatting.

I believe the authors acquired this dataset themselves thus more details of the bamboo dataset should be explained e.g preprocessing etc to enable other researchers to replicate their findings. Also, they should make the dataset publicly available for other researchers.

The whole manuscript needs minor grammatical correction

The result from comparison with other deep learning models should not be restricted to just ResNet and VGG, Kindly include more comparisons to verify your claims. DenseNet, Xception, Etc

More analysis should be carried out such as the AP and ROC performance which we help us know the class performance of the models. Sensitivity and Specificity

What are the limitation of this work? And the future direction

 Reference should be improved

Reviewer 3 Report

The paper has described an image recognition framework specifically for bamboo type classification. Specifically, they have incorporated residual module from the well-known ResNet CNN architectures into a transformer to increase the latter’s trainability on smaller dataset.

1-     The paper is generally written well except for a few typos and grammatical mistakes. E.g.

a.      “Image amount” should be “image count”

b.      Table 3 caption mentions F1 score but it is not provided

2-     Given that the major problem at hand is training a complex architecture (e.g. ViT) on a smaller dataset, the well-know solution in this regard is transfer learning or fine tuning the existing models to adapt to newer datasets especially of smaller sizes. However, the authors state that they didn’t consider fine tuning at all. Please provide an explanation for this approach. Multiple efforts have been reported in the literature which have handled this problem this problem earlier using transfer learning. E.g.

a.      S. Lee, S. Lee and B. C. Song, "Improving Vision Transformers to Learn Small-Size Dataset From Scratch," in IEEE Access, vol. 10, pp. 123212-123224, 2022, doi: 10.1109/ACCESS.2022.3224044.

b.      S. Park, B. -K. Kim and S. -Y. Dong, "Self-Supervised RGB-NIR Fusion Video Vision Transformer Framework for rPPG Estimation," in IEEE Transactions on Instrumentation and Measurement, vol. 71, pp. 1-10, 2022, Art no. 5024910, doi: 10.1109/TIM.2022.3217867.

c.      M. Bi, M. Wang, Z. Li and D. Hong, "Vision Transformer With Contrastive Learning for Remote Sensing Image Scene Classification," in IEEE Journal of Selected Topics in Applied Earth Observations and Remote Sensing, vol. 16, pp. 738-749, 2023, doi: 10.1109/JSTARS.2022.3230835.

3-     In continuation of the above comment, it would be advisable to include results for ResNet-18, ResNet-50 etc. trained using fine tuning (transfer learning) in Fig. 11 and Table 4 for a fair comparison.

4-     Fig. 6 implies a heavily imbalanced dataset. Have the authors considered using data augmentation to resolve this and the small size of the dataset? Data augmentation is a well known technique to address such issues.

Round 2

Reviewer 3 Report

The authors have adequately addressed all the concerns raised in the previous review cycle.